# Morphometric and Nanomechanical Features of Erythrocytes Characteristic of Early Pregnancy Loss

**DOI:** 10.3390/ijms23094512

**Published:** 2022-04-19

**Authors:** Ariana Langari, Velichka Strijkova, Regina Komsa-Penkova, Avgustina Danailova, Sashka Krumova, Stefka G. Taneva, Ina Giosheva, Emil Gartchev, Kamelia Kercheva, Alexey Savov, Svetla Todinova

**Affiliations:** 1Institute of Biophysics and Biomedical Engineering, Bulgarian Academy of Sciences, “Acad. G. Bontchev” Str. 21, 1113 Sofia, Bulgaria; arianalangari@abv.bg (A.L.); vily_strij@abv.bg (V.S.); avgustina_danailova@abv.bg (A.D.); sashka.b.krumova@gmail.com (S.K.); sgtaneva@gmail.com (S.G.T.); ina_gi@abv.bg (I.G.); 2Institute of Optical Materials and Technologies “Acad. Yordan Malinovski”, Bulgarian Academy of Sciences, “Acad. G. Bontchev” Str. 109, 1113 Sofia, Bulgaria; 3Department of Biochemistry, Medical University—Pleven, Sv. Kliment Ohridski Str. 1, 5800 Pleven, Bulgaria; rkomsa@gmail.com; 4University Hospital of Obstetrics and Gynecology “Maichin Dom”, Medical University Sofia, Zdrave Str. 2, 1431 Sofia, Bulgaria; egartt@gmail.com (E.G.); camellia.kercheva@gmail.com (K.K.); alexey.savov@abv.bg (A.S.)

**Keywords:** erythrocytes, early pregnancy loss, atomic force microscopy, cells’ senescence, membrane roughness, Young’s modulus, oxidative stress

## Abstract

Early pregnancy loss (EPL) is estimated to be between 15 and 20% of all adverse pregnancies. Approximately, half of EPL cases have no identifiable cause. Herein, we apply atomic force microscopy to evaluate the alteration of morphology and nanomechanics of erythrocytes from women with EPL with unknown etiology, as compared to healthy pregnant (PC) and nonpregnant women (NPC). Freshly isolated erythrocytes from women with EPL differ in both the roughness value (4.6 ± 0.3 nm, *p* < 0.05), and Young’s modulus (2.54 ± 0.6 MPa, *p* < 0.01) compared to the values for NPC (3.8 ± 0.4 nm and 0.94 ± 0.2 MPa, respectively) and PC (3.3 ± 0.2 nm and 1.12 ± 0.3 MPa, respectively). Moreover, we find a time-dependent trend for the reduction of the cells’ morphometric parameters (cells size and surface roughness) and the membrane elasticity—much faster for EPL than for the two control groups. The accelerated aging of EPL erythrocytes is expressed in faster morphological shape transformation and earlier occurrence of spiculated and spherical-shaped cells, reduced membrane roughness and elasticity with aging evolution. Oxidative stress in vitro contributed to the morphological cells’ changes observed for EPL senescent erythrocytes. The ultrastructural characteristics of cells derived from women with miscarriages show potential as a supplementary mark for a pathological state.

## 1. Introduction

Early pregnancy loss (EPL) or spontaneous abortion is the most common complication of human reproduction, with an incidence ranging between 50–70% of all conceptions [1]. EPL is usually associated with genetic or chromosomal abnormalities in the developing embryo [2], viral infections [3,4], immunological and immunogenic causes [5,6], thrombophilia [7], endocrine disorders [8], poorly controlled type 1 diabetes [9] and obesity [10]. However, nearly 50% of the cases are idiopathic (due to unknown etiology). Recent studies highlight the role of systemic and placental oxidative stress as an essential factor in the etiology of early pregnancy losses. In the early stages of pregnancy, trophoblast invasion occurs under low oxygen conditions creating a hypoxemic environment [11]. Trophoblasts are specialized cells of the placenta that play an important role in embryo implantation and interaction with the decidualized uterus. Hypoxia-induced oxidative stress is an important regulator of trophoblast invasion and is a part of the physiological development of the placenta and embryo in early pregnancy. When oxidative stress exceeds the normal physiological level, the complications such as miscarriage, preeclampsia [12], and limited intrauterine development [13] may occur.

Normal pregnancy involves profound alterations in various aspects of maternal hemostasis, such as imbalance in reactive oxygen species (ROS) formation, domination of the procoagulant effects over fibrinolytic activity, etc. These changes aim to maintain the placental function during pregnancy, but in some cases lead to a hypercoagulable state which is a prerequisite for thrombotic events. They can cause placental hypercoagulation leading to the microthrombi formation in the uteroplacental blood vessels, contributing to a hypoxic environment and, as consequence, to EPL in the first trimester along with other pathology [14,15,16]. Since erythrocytes bind large amounts of oxygen, they are one of the most exposed cells to oxidative stress in the body and are under the constant influence of free radicals such as superoxide and hydrogen peroxide (H_2_O_2_). Oxidative stress induces damage and deformation of the erythrocyte membrane and ultimately leads to faster cell aging and early eryptosis [17,18].

Despite the common belief that erythrocytes, or red blood cells (RBCs) play a passive role in hemostasis, there is growing evidence in the literature revealing their biological significance in blood clotting and/or blood dysfunctions. Erythrocytes are involved in thrombosis through their biochemical, biomechanical, and rheological properties, through the release of procoagulant signaling molecules or intercellular interactions. Many studies suggest that there is a relationship between RBCs abnormalities, such as elevated hematocrit [19], shape [20,21], or deformability [22], and thrombosis. Any defects in their structure underlie multiple hemolytic disorders [23]. Hence, the morphology and nanomechanics of erythrocytes may contribute to clarifying the etiology of some pathological situations.

Erythrocyte aging also affects hemostasis. It has been established that aged erythrocytes change their shape and surface area, which in turn affects blood viscosity. Senescent erythrocytes have been shown to have increased adhesion to endothelial cells due to the reorganization of their membrane [24].

Morphologic anomalies of the RBCs during pregnancy are poorly known. A minor alteration was observed but without clinical significance. Spherocytes or schistocytes can occasionally be found, even in the absence of hemolysis [25].

Abnormalities during pregnancy are largely associated with dysfunction of hemostasis and oxidative stress; therefore, the efforts of many researchers have focused on identifying potential biomarkers that would predict the risk of these complications. The introduction of new, non-invasive approaches that could complement current diagnostic tests would increase the accuracy of early identification of adverse pregnancies.

Atomic force microscopy (AFM), a high-resolution imaging technique, has been widely used for the characterization of the mechanical, morphological, and structural properties of red blood cells as well as the measurement of cell deformability in terms of Young’s modulus and membrane surface roughness [26,27,28,29].

The present work is based on the hypothesis that the morphology and nanomechanics of erythrocytes depend on various pathological factors and could serve as a marker for their presence. To validate or refute this hypothesis, changes in the topology and membrane elasticity of erythrocytes obtained from women with EPL and healthy controls (pregnant and nonpregnant controls) during cells aging are monitored by means of AFM and optical microscopy.

We reveal specific features of EPL erythrocytes for both young and senescent cells compared to those of the two control groups and a different pathway of morphological alterations during the RBCs aging process. The obtained data complement our understanding of the factors contributing to adverse pregnancy.

## 2. Results

### 2.1. Characteristics of the Groups of Women Included in the Investigation

The main clinical characteristics and hematological indices of erythrocytes derived from nonpregnant (NPC) and pregnant (PC) controls and patients with EPL are presented in Table 1. The basic hematological parameters for the three studied groups were similar and did not deviate from the reference values.

### 2.2. Changes in Erythrocytes’ Shape during Aging

After their isolation, the erythrocytes were monitored for a period of 40 days by means of optical microscopy and AFM. The aging of erythrocytes from EPL, PC, and NPC is associated with shape transformation from the typical for young cells and was classified into four different morphological types designated as biconcave; crenated; spiculated, and spherocytes (Figure 1) as already described [30].

The quantitative determination of the percentage of each morphological class was performed using an optical microscope (for a typical image see Appendix A). More than 30 images were taken and analyzed for every test sample, and the erythrocytes were classified into one of the four morphological types. The results are presented as the histograms for the 1st, 10th, 20th, and 30th days after the initial isolation of the cells (Figure 2). Data analysis showed that the morphology of freshly isolated erythrocytes did not differ significantly between the two control groups, NPC and PC. The biconcave cells were the dominant form (75% and 68% for NPC and PC, respectively, *p* > 0.05, Figure 2A), while the remaining ones had a crenated shape (Figure 2B), i.e., no other morphological types were observed in freshly isolated “healthy” erythrocytes. Although the biconcave cells prevailed in the EPL group, their proportion was lower (53%, *p* < 0.05) compared to the control groups. A small population of spiculocytes and spherocytes was also recorded for the EPL group (Figure 2C,D).

During the aging path, the distribution of erythrocyte types changed. The number of biconcave cells gradually decreased at the expense of other morphological types. However, the trend of this transformation was different for the studied groups, i.e., slower for the two control groups (NPC and PC) as compared to that of women with EPL. For NPC and PC groups, on day 10 of the follow-up study, the biconcave and crenated shape were reduced by nearly 20% but were still dominant. About 10% of spiculated cells were also found in this study group (Figure 2C). On day 20, the biconcave and crenated cells had approximately the same distribution (*p* > 0.05), and an increase of the other two forms characteristic of senescent cells (spiculocytes and spherocytes) was observed. At the end of the follow-up, the spiculocytes and spherocytes were prevalent (Figure 2C,D). In the course of storage time, the transformation of EPL erythrocytes was evidently much faster than in the two control sets, reflecting the accelerated cell aging process (Figure 2). Along the aging path, the portion of erythrocytes with spiculed and spherical shapes augmented significantly, and on day 20, were the dominant morphological EPL type (Figure 2C,D).

The nanoscale morphology of RBCs was further examined using AFM. Figure 3 displays selected AFM images showing the change over time in the morphological erythrocyte classes obtained from the three study groups. It can clearly be seen that the different morphological classes characteristic of senescent erythrocytes such as spiculocytes and spherocytes appear earlier in the EPL group (Figure 3H,I) than in NPC and PC ones (Figure 3A–F). In controls, the biconcave and crenated cells were still detected on day 30 of the follow-up time (Figure 3C,F), while for the EPL group, these cell types were not observed.

### 2.3. Morphometric Characteristics of Erythrocytes in the Course of Their Aging

The diameter of erythrocytes is a valuable parameter of health status [31]. In the course of RBCs aging, the value of this parameter decreased. The average diameter of RBCs on the first day of cell isolation was about 7.3 µm and decreased along with aging to about 5.7 µm on day 30 and remained constant at the end of the storage time for all groups under study but followed different patterns for each group (Figure 4). The diameter of NPCs and PCs gradually decreased until the end of the storage period, reaching the minimum value (Figure 4), the former having slightly higher values (*p* > 0.05) up to the 30th day. The diameter of EPL cells decreased sharply by day 10 and changed slowly in the next 20-day period (Figure 4).

The change of the membrane roughness (R_rms_) of erythrocytes as a function of the storage time had similar behavior. The decreasing trend of R_rms_ for the NPC group was linear between the 1st (3.3 ± 0.2 nm) and 40th (1.3 ± 0.3 nm) day (Figure 5A). For the PC group, the decline was exponential, starting from a higher value for freshly isolated RBCs (3.8 ± 0.4 nm, *p* > 0.05) than in the NPC group (Figure 5A). The membrane roughness for freshly isolated EPL erythrocytes was significantly higher (4.6 ± 0.3 nm) compared to the two control groups (*p* < 0.05) and decreased dramatically, reaching a constant value of 1.2 ± 0.2 nm on the 20th day (Figure 5A).

We also studied how the membrane roughness alterations were correlated with the structural modifications occurring in cells membrane along the aging path. Figure 5B–J reveals the ultrastructural changes of erythrocytes’ membrane derived from the three studied groups. The membrane surface of healthy NPC and PC erythrocytes appeared homogeneous throughout the storage period with gradual smoothing of the structure, particularly at the later stage (Figure 5D,G). In contrast, the surface of fresh EPL erythrocytes was heterogeneous with irregularly dispersed small lumps (Figure 5H). An important feature of EPL erythrocytes was the faster occurrence of structural alterations and the appearance of microvesicles at later storage times (Figure 5I,J).

### 2.4. Young’s Modulus of Erythrocytes in the Course of Their Aging

Young’s modulus (Ea) is a reliable indicator of the mechanical state of blood cells membranes, i.e., their stiffness. Table 2 presents the average values of Ea for erythrocytes obtained from the studied groups. The stiffness of fresh NPC and PC erythrocytes did not differ statistically (*p* > 0.05), and they both increased gradually up to the 30th day of the storage time (Table 2, Figure 6) and remained constant for the next 10 days. Young’s modulus of fresh and senescent erythrocytes from EPL women was significantly higher than that of the control groups (*p* < 0.01), but the rate of increase over time was lower (Table 2, Figure 6). As in controls, no significant change in Ea value was observed after the 30th day.

### 2.5. Induced Oxidative Stress in Erythrocytes

In order to obtain further insight into the predisposition of erythrocytes to oxidative damage, we next examined the morphometric properties of newly isolated erythrocytes after their treatment with H_2_O_2_ solution (100 mM and 200 mM) mimicking the conditions of acute (short-term) oxidative stress. The relative abundance of the morphological type of freshly untreated erythrocytes was similar to those reported in Figure 2: the biconcave discs were the dominant ones (69%), followed by the crenated (26%) cells; a small percentage (5%) of spiculated RBCs were also observed (Figure 7A and Appendix A). Exposure to 100 mM H_2_O_2_ significantly reduced the cells with a biconcave shape at the expense of the crenated and spiculated ones (Figure 7B and Appendix A). With regard to the 200 mM H_2_O_2_ treatment, the spiculocytes become the dominant class while the biconcave type almost disappeared. The proportion of spherocytes was also significant (Figure 7C and Appendix A). The average roughness was 2.41 ± 0.1 nm and 2.0 ± 0.2 nm for 100 and 200 mM H_2_O_2_ treated cells, respectively, which is significantly lower (*p* < 0.05) compared to untreated cells (3.4 ± 0.3 nm).

We tested the effect of oxidative stress on erythrocyte morphology alteration and how this simulation can be related to the normal aging process of the cells. For this purpose, we compared the newly isolated erythrocytes from healthy controls treated with H_2_O_2_ (Figure 7) and senescent EPL erythrocytes (Figure 2). Although there is no complete coincidence between the morphological changes that occur in RBC of the EPL group with aging evolution and modification of freshly obtained cells of healthy controls after exposure to different H_2_O_2_ concentrations, some analogies can be made. A higher proportion of cells with reduced deformability and functionality and the decrease of membrane roughness (i.e., spiculocytes and spherocytes) upon raising hydrogen peroxide concentration (Figure 8A,B), can be compared to the morphological cells’ alteration of 10-day-old and 20-day-old EPL RBCs (Figure 8C,D and Appendix A).

## 3. Discussion

Erythrocytes are subjected to a constantly changing environment in the bloodstream that in turn can strongly alter their structural and biophysical properties, including cell geometry, membrane structure, and cytoskeleton flexibility [32]. In some pathologies (e.g., diabetes, sickle-cell disease, neuroacanthocytosis syndromes, cardiovascular, hemoglobinopathies, malaria, etc.), as well as in aging, the biochemical and biophysical characteristics of RBCs are drastically altered. Therefore, the characterization of the biophysical features of RBCs is important for the assessment of their functionality.

Our previous study on the thermodynamic behavior of erythrocytes demonstrated accelerated aging-related changes in the calorimetric profiles of RBCs in women with early pregnancy loss compared to healthy pregnant and nonpregnant controls [33]. Herein, we expanded our research by exploring the nanomechanical characteristics and ultrastructural changes occurring during the aging process of erythrocytes obtained from the same group.

From a morphological point of view, the normal aging of erythrocytes was visualized as cells’ shape transformations (from the normal biconcave disc through different morphological classes to complete loss of the normal shape, i.e., dense spherocytes). All these modifications were accompanied by a gradual decrease in the size and membrane roughness of the cells. The diameter reduction was associated with the shape change. We established that the diameter of biconcave and crenated cells was larger than that of spiculated and spherical RBCs. We also found different aging paths, more severity, and faster occurrence of morphological alterations of EPL erythrocytes compared to both control groups. The increase of spiculocyte and spherocyte populations occurred significantly earlier in the follow-up time in EPL women compared to the two control groups. The cell shape alteration due to the partial or complete loss of deformability may occur as a result of various pathological conditions [34], accumulation of non-neutralized reactive oxygen species (ROS), or as a part of RBC aging [35]. It is known that the shape of erythrocytes is strongly determined by the mechanical properties of the cytoskeleton and by adenosine triphosphate (ATP) abundance [36]. It was established that the aging path is triggered by the ATP intracellular concentration that influences the membrane-skeleton structure [28]. ATP is important for maintaining the discoid shape of RBC since the depletion of ATP causes a change from the discoid to the echinocyte shape [37]. For example, sickle RBCs are characterized by elevated indices of oxidative stress and depressed ATP [38]. Significantly lower erythrocyte ATP levels have been shown for pregnant than in nonpregnant women [39].

An important result of our study is the difference between the membrane roughness of fresh EPL erythrocytes and those of the two control groups. The significantly higher R_rms_ value detected for EPL is due to the more uneven folding of the membrane and the appearance of some protrusions probably as a consequence of the disturbed integrity of the plasma membrane and/or the beginning of microvesiculation. Vesiculation of mature RBCs contributes to the removal of defective patches of the erythrocyte membrane. It is known that in some pathological situations and in the RBCs aging process, vesiculation of the plasma membrane is intensified [40,41]. The release of microvesicles (microparticles) takes place as a result of the redistribution of phosphatidylserine and phosphatidylethanolamine from the inner to the outer leaflet of the plasma membrane and occurs selectively in lipid-rich microdomains within the plasma membrane [42]. Our previous study revealed age-related decreases in erythrocyte membrane roughness for healthy individuals [30]. Herein we also show a decrease in erythrocytes roughness with cells aging, with a drastically faster rate in EPL erythrocytes than in controls. It might be related to enhanced oxidative stress, especially in adverse pregnancy outcomes as was previously presented [43,44,45]. Due to the increased oxygen metabolism in the mother and fetus, pregnancy itself is associated with increased oxidative stress. However, overexposed oxidative stress can be the prerequisite for many reproductive complications including infertility, miscarriage, pre-eclampsia, fetal growth restriction, etc. [46]. Several studies demonstrated that systemic and placental oxidative stress is a major factor in the etiopathogenesis of early pregnancy loss [47,48]. Indeed, it was found that in placental cell cultures from women with early miscarriages, the level of stress in the structure of the endoplasmic reticulum is higher, and the response of endoplasmic reticulum chaperone proteins, which are a repair mechanism, is less effective [49]. Oxidative conditions can trigger RBC membrane proteins phosphorylation, mainly involving Band 3 protein [50]. Band 3 contributes significantly to maintaining the plasticity and integrity of the RBC membrane, through the attachment of an N-terminal cytoplasmic domain to cytoskeletal spectrin, through the mediator’s protein 4.2, and ankyrin. This interaction is essential for RBC morphology and stability. Oxidative damage of Band 3, leading up to accelerated RBC senescence, is associated with Band 3 clustering and enhanced avidity of autoantibody binding, thus causing the removal of erythrocytes from circulation [51]. The binding of denatured oxidized hemoglobin (hemichromes) to Band 3 triggers this clustering [52], which in turn leads to a weakening of the vertical connectivity with the spectrin-binding complex and ultimately reduces the membrane deformability. In line with this, McPherson et al. [53] have observed that crosslinking of Band 3 via hemichromes caused substantial immobilization of the membrane domain of Band 3.

The obtained data on the morphometric characteristics of RBCs are consistent with our previous results on the thermodynamic behavior of erythrocytes isolated from the same groups of women. The cross-correlation analysis of the calorimetric data obtained by means of differential scanning calorimetry [33] and AFM studies, shows a strong positive correlation between the destabilization of the transition of Band 3 protein (T_m_^band3^), determined by the thermograms of the studied groups, and the corresponding value of the membrane roughness. The Pearson correlation coefficient in both control groups is r = 0.93 (NPC) and r = 0.91 (PC), respectively, which is in line with our previous finding that structural changes in the cell membrane directly affect the thermodynamic characteristics of erythrocytes [30]. For the EPL group, a lower value of r (r = 0.87) is obtained, most likely due to the fact that the temperature-induced transition of Band 3 protein denaturation was not observed after the 20th day of cell isolation [33] and the statistical analysis was performed for a shorter period.

Our data also show strongly reduced membrane elasticity of EPL erythrocytes relative to that of the controls for both fresh and senescent cells. The erythrocyte membrane elasticity has been regarded as one of the most useful markers of health status. Its quantitative determination is crucial for understanding the mechanisms leading to the impairment of the RBC deformability. Decreased elasticity is a sign of reduced cell lifespan [54] and can be caused by various factors related to oxidative stress [55], altered membrane composition, and pathological conditions. For example, the ratio between phospholipids and cholesterol content has been reported to influence red blood cell mechanical properties and lead to an increase in membrane rigidity in diabetes mellitus samples [56]. Increased membrane stiffness is characteristic of erythrocytes derived from patients with neurodegenerative diseases compared to healthy ones [57]. Altered RBC deformability can be related to some diseases (such as sickle cell disease) or to abnormal membrane protein phosphorylation or RBC vesiculation.

The established negative correlation between Young’s modulus and R_rms_ value of the erythrocyte membrane (Appendix A) is in support of the hypothesis that the reorganization of the erythrocyte membrane is largely related to the clustering of the membrane domain of Band 3 protein and disruption of the membrane-cytoskeletal junction of cells. All these membrane alterations lead to significant modifications in the nanomechanical properties of the plasma membrane.

Erythrocyte aging is also accompanied by the inactivation of cellular enzymes (including antioxidant protection enzymes) and many membrane transporters, which reduces the neutralization of reactive oxygen species (ROS) [17,58]. In order to determine the effects of ROS on the morphological cells’ transformation, we simulated oxidative stress induced by hydrogen peroxide on freshly isolated erythrocytes. Although this model cannot accurately represent the state of accelerated aging of red blood cells, it sheds some light on the factors that may affect it. Oxidation of RBCs with hydrogen peroxide leads to a significant reduction in cells with normal deformability and functionality (i.e., biconcave and crenated RBCs) at the expense of those with reduced functionality (i.e., spiculocytes and spherocytes). These alterations mimic the morphological changes that occur in EPL with the evolution of cells’ aging. The results suggest that the protective mechanisms against ROS are likely to be weakened or damaged in the RBCs in the EPL group compared to the controls. Mohanty et al., demonstrated that the increase in the affinity of partially oxygenated hemoglobin to the RBC membrane limits the efficiency of the antioxidant system [17]. As was mentioned above, early pregnancy losses are associated with increased oxidative stress, while in a successful pregnancy, the availability of antioxidants in the peripheral blood such as blood-plasma thiol and ceruloplasmin protects from oxidative attack [59].

## 4. Materials and Methods

### 4.1. Selection of Women with EPL and Healthy Controls

Twenty-six women (mean age 34 ± 6 (19 to 40 years)) suffering first-trimester EPL and healthy pregnant, and nonpregnant women were enrolled in the study after signing informed consent. The study is approved by the Ethics Committee of Medical University-Pleven (approval no. 404-KENID 22/10/15) and was performed in accordance with the Helsinki international ethical standards on human experimentation.

The gestational weeks of EPL women were 8.2 ± 1.4 (range 6–11) at the time of the abortion. Women with obstetric and endocrine disorders such as inflammatory changes in the internal genital organs, carrier of balanced mutations (not occurring in the early period of pregnancy), or endocrine disorders were excluded from the selection. The two control groups consist of 10 healthy, nonpregnant women (mean age 36 ± 7 (23 to 41 years)), age-matched to the study group of patients with one or more live births, and eight pregnant females (mean age 28 ± 3 (25 to 35 years)), without a history of previous early pregnancy loss and without pathologies.

### 4.2. Erythrocyte Preparation

Blood was collected in EDTA tubes vacutainers (0.084 mL 15% EDTA Becton, Dickinson, and Company, Franklin Lakes, NJ, USA), washed in PBS buffer (140 mM NaCl, 2.7 mM KLC, 8 mM Na_2_HPO_4_, 1 mM KH_2_PO_4_, and 1 mM EDTA), pH 7.4, and centrifuged three times (1200× *g*, 15 min, at 4 °C) to remove the plasma and leukocytes. After washing, erythrocytes were diluted to 30% hematocrit in PBS solution and stored at 4 °C for analysis. The process of washing erythrocytes was repeated before each experiment. Smears of erythrocyte suspension diluted with blood plasma (in a ratio of 1:1) were made according to the procedure described in Dinarelli et al. [60]. The cells deposited on the poly-L-lysine slides were further observed using an optical and atomic force microscope (AFM).

### 4.3. Optical Microscopy

Modern optical microscopes are able to magnify an object 1500 times and are increasingly used in clinical applications [61,62]. The morphological types of RBCs isolated from the control groups and EPL women were determined from the optical images of the cells obtained by optical microscope (3D Optical profiler, Zeta-20, Zeta Instruments, Milpitas, CA, USA). All images were captured using a magnification 50× objective lens. The concentration of RBCs in the sample is low enough to prevent a significant number of them from attaching to each other. Typically, the number of analyzed cells per sample was about 350. The experiments were performed at room temperature.

### 4.4. Atomic Force Microscopy Measurement

AFM measurements were performed using Atomic Force Microscope (MFP-3D, Asylum Research, Oxford Instruments, Abingdon, UK) operated in contact mode at room temperature. Silicon nitride probes—Nanosensors, type qp-Bio, with a spring constant of 0.06 N/m, a resonant frequency of 16 kHz, and a nominal tip radius of 8 nm (the shape is conical) were employed in all AFM measurements. Determination of morphometric parameters (erythrocytes’ diameter, membrane roughness) was carried out using Gwyddion and IgorPro 6.37 software. The roughness analysis was performed in an area of 2.0 × 2.0 µm in the central part of the erythrocytes. Before each measurement, a first-order flattening was applied to avoid the effect of cell curvature. The calculation of membrane roughness was performed as the mean square value of the heights of the surface [27]
(1)$$R_{\mathrm{rms}}=\sqrt{\sum_{i=1}^{N}\frac{{\left(z_{i}-z_{m}\right)}^{2}}{\left(N-1\right)}}$$
where *N* is the total number of points, *z_i_* is the height of the *i*th point and *z_m_* is the mean height.

The elastic properties of the cells were described by Young’s modulus (Ea) determined by the force-distance (f-d) curves. The mapping of f-d curves was preceded by type calibration on a clean bare glass using special IgorPro software, embedded in the AFM system [63,64]. The cells from each group were analyzed by selecting a 1 µm-by-1 µm scan area on the periphery of the biconcave and crenated RBCs and on the central part for the spiculocytes and spherocytes performing 32 by 32 data points of individual force curve measurements. The quantitative measurements of Young’s modulus were obtained by fitting the slope of any force-distance curve of the image to an appropriate model (in this instance, the Hertz model, refined by Briscoe for conical probe) [65]:(2)$$F(\delta )=\frac{2E\left(\tan\theta \right)}{\pi \left(1-\nu^{2}\right)}$$
where F is the loading force, *δ* is the indentation depth, *E* is the apparent Young’s modulus, 2*θ* is the open-angle of the type, and *ν* is Poisson’s ratio. The Poisson’s ratio of the cell is set as 0.5 in this study for its incompressibility.

### 4.5. RBC Oxidation Procedure

The oxidation of erythrocytes was carried out by the modified method of Hale 2011 [66]. To compare the effect of the two-oxidant concentration, bulk concentrations of 100 mM and 200 mM were used. H_2_O_2_ solutions in PBS were prepared from fresh stock solution (1 M) immediately before each experiment to minimize peroxide degradation. To disperse the necessary concentration into PBS, the mixture was vigorously shaken. Blood samples (6 mL) were taken from three volunteers. The erythrocytes isolation was performed according to the procedure described in paragraph 4.2. A suspension of isolated RBCs (600 µL) was incubated with 200 µL of 100 mM and 200 mM H_2_O_2_ correspondingly (H1009 Sigma-Aldrich Pty Ltd, an affiliate of Merck KGaA, Darmstadt, Germany) for 4 h at room temperature. The reaction was stopped with 200 µL 10 mM EDTA. The untreated and H_2_O_2_ treated cells were deposited on poly-L-lysine slides for further observation using optical microscopy and AFM.

### 4.6. Statistical Analysis

All data were expressed as means ± SD (standard deviation). Statistically significant differences between means were evaluated by a non-parametric Wilcoxon test [67] and were assumed significant at *p* < 0.05. Linear correlation between parameters was tested by Pearson’s correlation coefficient r [68].

## 5. Conclusions

For the first time, specific morphometric characteristics of erythrocytes from women with early pregnancy loss were identified, distinguishing them from those of healthy pregnant and nonpregnant women. The study reveals substantial differences in the membrane ultrastructure and elasticity of fresh EPL erythrocytes compared to those of NPC and PC. Therefore, the roughness and Young’s modulus values of freshly isolated erythrocytes can be considered indicators of a pathological condition in pregnancy. Data analysis also provides evidence for accelerated aging of EPL RBCs, characterized by: (i) acceleration in morphological shape transformation and faster occurrence of spiculated and spherical-shaped cells; (ii) different patterns of the reduction of cells’ size and membrane roughness and (iii) enhanced stiffness with aging evolution.

The results of the simulated oxidative stress by hydrogen peroxide in freshly isolated erythrocytes suggest that the antioxidative defense is likely to be weakened in RBCs of women with EPL compared to healthy controls.

The obtained data might serve as the basis for the development of a morphometric algorithm for the distinction of different pathological conditions during pregnancy.

## Figures and Tables

**Figure 1 ijms-23-04512-f001:**
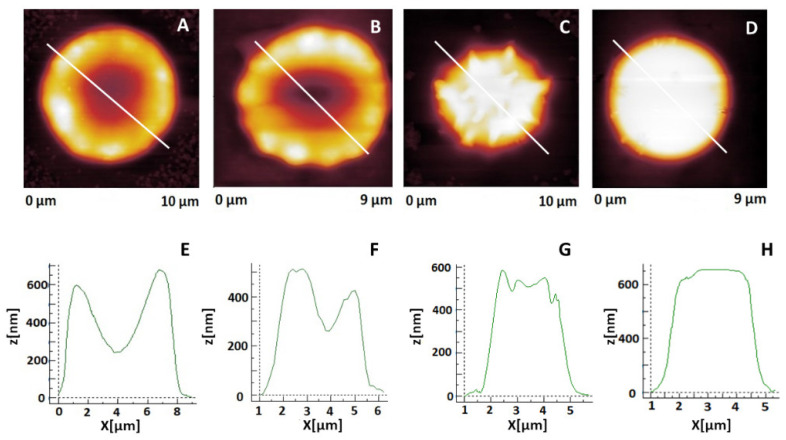
Typical AFM (2D) images of the four morphological classes of erythrocytes during cells’ aging: (**A**) biconcave disk; (**B**) crenated; (**C**) spiculated, and (**D**) spherocytes. The corresponding cross-sectional profiles of the cells defined by the white line of the AFM images from panels (**A**–**D**) are presented in panels (**E**–**H**), respectively.

**Figure 2 ijms-23-04512-f002:**
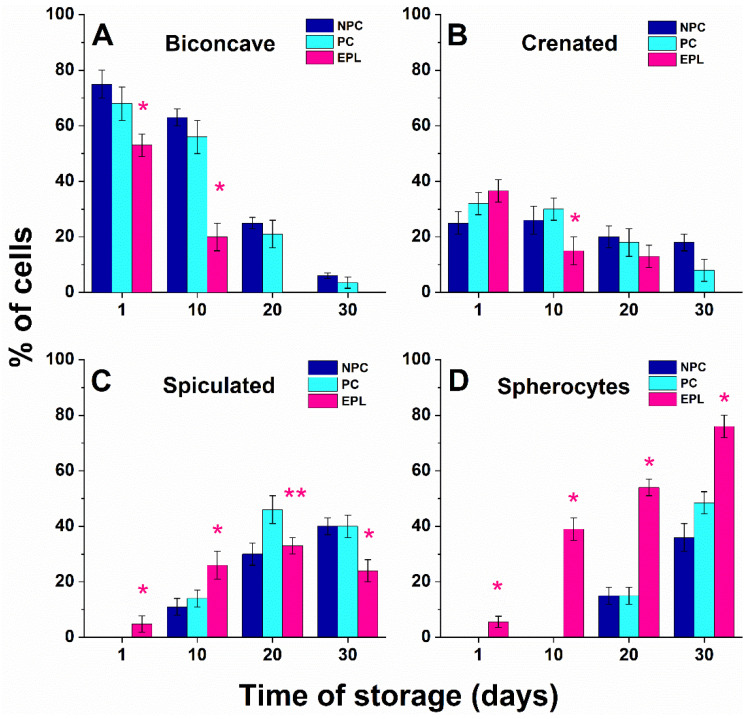
Histogram of the relative fraction of erythrocytes assigned to the four morphological types (**A**–**D**) during aging of the cells isolated from the blood of healthy nonpregnant controls (NPC, dark blue bars), healthy pregnant controls (PC, light blue bars), and women with early pregnancy loss (EPL, red bars). Each morphological type is represented as a percentage of the total number of cells. Non-parametric Wilcoxon test was used to determine statistical differences in the fraction of different morphological types of EPL cells, which are given as *p* values, indicated as * *p* < 0.05 relative to both NPC and PC, ** *p* < 0.05 relative to PC.

**Figure 3 ijms-23-04512-f003:**
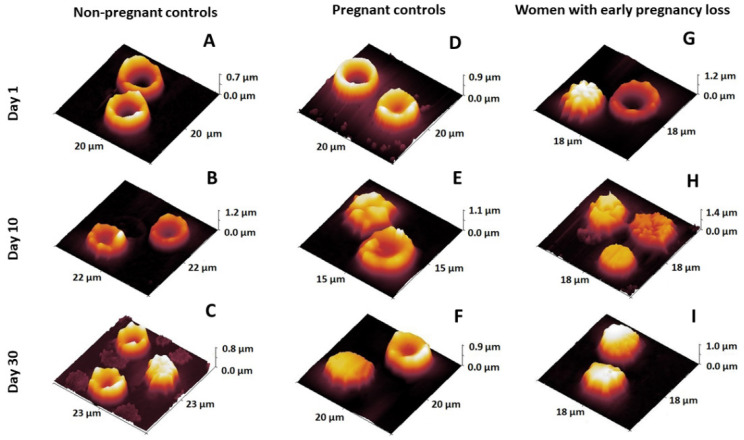
Selected 3D images of erythrocytes isolated from the studied groups of women (**A**–**I**) presenting morphological changes in the process of cell aging.

**Figure 4 ijms-23-04512-f004:**
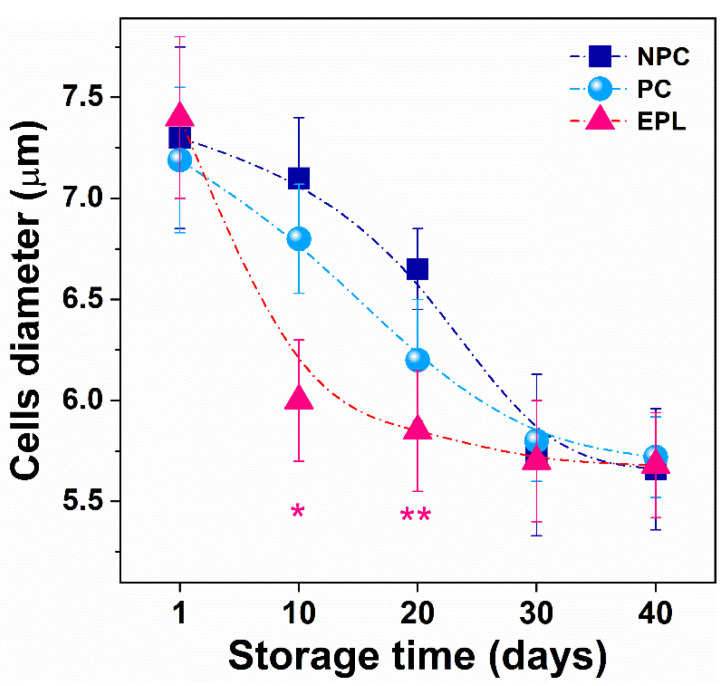
Alteration in the erythrocyte’s diameter determined for cells isolated from nonpregnant controls (NPC, dark blue square), pregnant controls (PC, light blue sphere), and women with early pregnancy loss (EPL, red triangle) as a function of the aging time of cells. Mean values and SD. Non-parametric Wilcoxon test was used to determine statistical differences in the value of the diameter of EPL cells, which are given as *p* values, indicated as * *p* < 0.05 relative to both NPC and PC, ** *p* < 0.05 relative to NPC.

**Figure 5 ijms-23-04512-f005:**
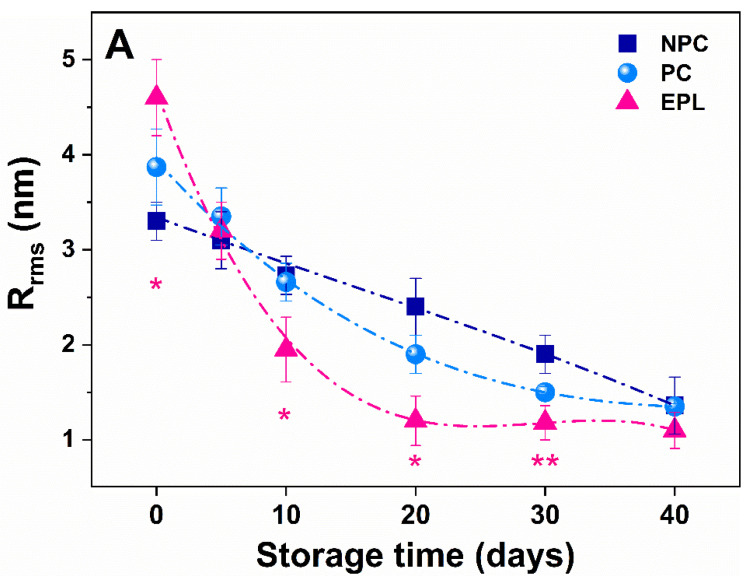
The calculated value of R_rms_ of the plasma membrane as a function of cell storage time (**A**) and the ultrastructural changes over time of erythrocytes membranes of cells from nonpregnant controls (NPC) (**B**–**D**); pregnant controls (PC) (**E**–**G**); and women with early pregnancy loss (EPL) (**H**–**J**). The scale bars indicate 500 nm. Non-parametric Wilcoxon test was used to determine statistical differences in R_rms_ value of EPL cells, which are given as *p* values, indicated as * *p* < 0.05 relative to both NPC and PC, ** *p* < 0.05 relative to NPC.

**Figure 6 ijms-23-04512-f006:**
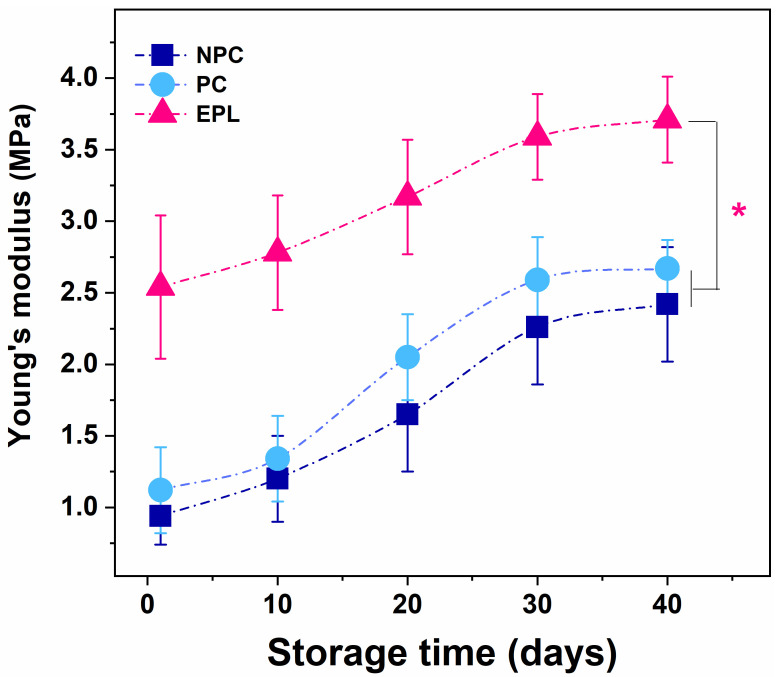
Young’s modulus values of erythrocytes aging recorded on the 1st, 10th, 20th 30th, and 40th isolate from nonpregnant controls (NPC), pregnant controls (PC), and women with early pregnancy loss (EPL). Non-parametric Wilcoxon test was used to determine statistical differences, which are given as *p* values, indicated as * *p* < 0.01 relative to both NPC and PC.

**Figure 7 ijms-23-04512-f007:**
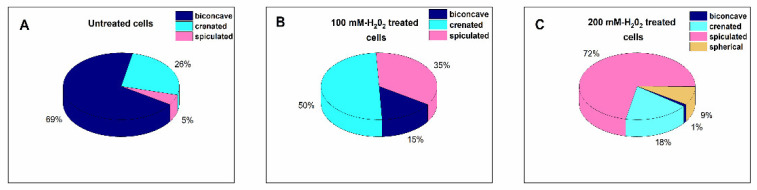
Erythrocytes’ morphology classes obtained by means of optical microscopy determined for newly isolated untreated cells from healthy controls (**A**) and cells subjected to 100 mM (**B**) and 200 mM (**C**) H_2_O_2_ treatment. Data are presented as percentages of the total number of cells.

**Figure 8 ijms-23-04512-f008:**
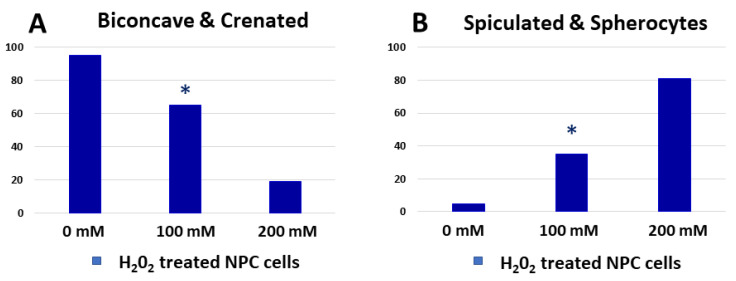
Comparison between the percentages of the morphological types of newly isolated cells from healthy controls treated with H_2_O_2_ (**A**,**B**) and EPL erythrocytes 10-day-old and 20-day-old (**C**,**D**). Non-parametric Wilcoxon test was applied to determine statistical differences in the value of the fraction of different morphological types of H_2_O_2_ treated cells, which are given as *p* values, indicated as * *p* < 0.05 relative to the respective morphological fractions of 10-day-aged EPL erythrocytes.

**Table 1 ijms-23-04512-t001:** RBCs indices: mean corpuscular volume (MCV), mean corpuscular hemoglobin (MCH), mean corpuscular hemoglobin concentration (MCHC), and red cell distribution width (RDW); age (mean and SD) of nonpregnant controls (NPC), pregnant controls (PC), and women with early pregnancy loss (EPL) and the gestational weeks (GW) of women with EPL.

Subjects	Age (Years)	GW at the Time of Abortion	MCV (mmol·L^−1^)	MCH (pg·L^−1^)	MCHC (g·L^−1^)	RDW (%)
NPC	36 ± 7		94.0 ± 7.0	30.0 ± 3.0	330 ± 14	12.5 ± 1.3
PC	28 ± 3		97.8 ± 0.1	32.5 ± 0.4	333 ± 4	12.8± 0.6
EPL	34 ± 6	8.2 ± 1.4	89.3 ± 5.4	30.6 ± 2.0	342 ± 15	12.8 ± 1.6

**Table 2 ijms-23-04512-t002:** Young’s modulus was calculated for fresh and aged erythrocytes derived from nonpregnant controls (NPC), pregnant controls (PC), and women with early pregnancy loss (EPL).

Groups	Ea (MPa)				
	Day 1	Day 10	Day 20	Day 30	Day 40
NPC	0.94 ± 0.2	1.20 ± 0.3	1.65 ± 0.5	2.26 ± 0.5	2.42 ± 0.4
PC	1.12 ± 0.3	1.34 ± 0.3	2.05 ± 0.5	2.59 ± 0.5	2.67 ± 0.4
EPL	2.54 ± 0.6 *	2.78 ± 0.5 *	3.17 ± 0.4 *	3.59 ± 0.4 *	3.71 ± 0.3 *

* Indicates statistically significant difference (*p* < 0.01) from the respective NPC control value.

## Data Availability

The data are contained within the article or the Appendix A.

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
