# Peer review of "Morphometric and Nanomechanical Features of Erythrocytes Characteristic of Early Pregnancy Loss"

_ijms, 2022, doi:10.3390/ijms23094512_

Round 1
Reviewer 1 Report
Dear Authors, the present manuscript is great work. Only few minor improvements are required.
For example, if I understood correctly, 26 EPL women while only 10 were inrolled in the study as controls. How much does this effect the uncertainty of the comparison?
The age of the subjects in NPC and EPL was remarkably higher compared to PC. How much is this distorting the results? In some aspects only the NPC anc EPL look similar while both dissimilar to PC. COuld this have any connection to cohort age?
On what criteria were discriminated the biconcave disks from the crenated ones? Some explanation in this sense would ease the understanding.
Fig4 shows the cell diameter depending on storage time. WHat is the reason behind the fact that all groups end up at the same point at day 30? (Same question for Rrms at Fig 5 with the difference that they catch up at day 40. Why was cell diameter studied for shorter period than others?)
Fig 8 compares the effect of H2O2 on cell morphology, however it is not entirely clear which is the basis of comparison, the time or H2O2 concentration. It seems that both increase which makes impossible to decide which effect is presented.
Please specify all components in formula 2. Theta is missing form the text. For AFM experts it might be familiar but for larger audience might not.
Minor spell check is required, like CNP is written instead NPC.
Author Response
Reviewer 1
Dear Authors, the present manuscript is great work. Only few minor improvements are required.
Dear Reviewer,
We deeply appreciate your very insightful and constructive comments and recommendations. Hereby is a point-by-point reply to your comments. Changes made to the manuscript have been marked with track changes.
For example, if I understood correctly, 26 EPL women while only 10 were enrolled in the study as controls. How much does this effect the uncertainty of the comparison?
Answer: We agree with the reviewer that a larger control group would provide more significant statistical data. However, due to personal reasons and the Covid situation six of the controls withdrawed from the study.
We use a statistical power analysis to estimate the minimum sample size required for our experiment for desired significance level. The control of 10 subjects is completely satisfying for the non-parametric calculations of the morphology and morphometric parameters, as well as the modulus of elasticity for the isolated cells.
The age of the subjects in NPC and EPL was remarkably higher compared to PC. How much is this distorting the results? In some aspects only the NPC anc EPL look similar while both dissimilar to PC. COuld this have any connection to cohort age?
Answer:
- The main idea when selecting the groups was to have a similar age of EPL and NPC. Women with EPL experience miscarriage and are trying to conceive several times, they are usually elder. This is the reason that the pregnant women cohort is generally younger.
- Here we step on the article investigating the changes in the RBC membrane with cells age including a large number of subjects. They did not find a difference in the distribution of membrane proteins complement decay‑accelerating factor (CD55) and CD59 glycoprotein precursor (CD59) exposition in the groups from 16-32 years and older than >32 years.
John N. Waitumbi, Béatrice Donvito, Aymric Kisserli, Jacques H. M. Cohen, Joseé A. Stoute, Age-Related Changes in Red Blood Cell Complement Regulatory Proteins and Susceptibility to Severe Malaria, The Journal of Infectious Diseases, Volume 190, Issue 6, 15 September 2004, Pages 1183–1191, https://doi.org/10.1086/423140
On what criteria were discriminated the biconcave disks from the crenated ones? Some explanation in this sense would ease the understanding.
Answer: Both the biconcave and crenated cells are young erythrocytes. In this regard, they have almost similar parameters. However, in contrast to biconcave cells, the crenated ones have lost the shape of a regular disk, and some protrusions (precursors of spicules) appear on the periphery are clearly visible (for clarity, see the figure below and Figure S1 (A and B) of Supplementary). The crenated RBCs can be considered as the intermediate form before RBCs transformation into spiculocytes in the course of the aging path.
Fig4 shows the cell diameter depending on storage time. WHat is the reason behind the fact that all groups end up at the same point at day 30? (Same question for Rrms at Fig 5 with the difference that they catch up at day 40. Why was cell diameter studied for shorter period than others?)
Answer: We thank the reviewer for this comment!
The diameter of erythrocytes strongly depends on the cell’s shape, i.e., the young erythrocytes (the biconcave and crenated ones) are larger than the senescent cells. Our results show a different aging pathway for the study groups. However, at the end of the follow-up period, the populations of spiculocytes and spherocytes were dominant for the three groups, and hence the mean cell size was similar. Similar explanation can be applied regarding the roughness value. In healthy and young RBCs higher roughness values are obtained when compared to senescent or to erythrocytes affected by membrane-skeleton pathologies. This can be related either to permanent damage to the skeletal architecture, as in the case of some pathologies (mentioned in the ‘Discussion” section), or to weakening of the contacts between the membrane and the cytoskeleton, as in the case of cell aging. As the EPL patients do not have concomitant diseases (such as diabetes, spherocytosis, etc.), the final value of the RBC parameters must be due to the cells’ aging process.
We agree with the reviewer that all parameters should be shown for the same period of time and therefore in the revised version we present data up to 40th day.
Fig 8 compares the effect of H2O2 on cell morphology, however it is not entirely clear which is the basis of comparison, the time or H2O2 concentration. It seems that both increase which makes impossible to decide which effect is presented.
Answer: We tested the hypothesis on oxidative stress contribution to erythrocyte morphology alteration in EPL patients. We incubated freshly isolated RBC from healthy controls with hydrogen peroxide aiming to simulate oxidative stress conditions as a possible mechanism causing a complex behavior of cells during their aging.
The hypothesis that the erythrocytes derived from patients with early pregnancy loss are with disturbed oxidation-anty-oxidation balance in favor of oxidants, can eventually lead to damage of red blood cells.
We compared the morphological types of fresh erythrocytes isolated from healthy controls after H2O2 treatment with the erythrocytes from the experiment on RBC aging reported in Fig.2.
As a result of this investigation, we found that the change in morphological types in cells treated with H2O2 (100 mM and 200 mM) was close to that of 10-day-old and 20-day-old EPL erythrocytes illustrated on Fig. 8.
Here we prove that H2O2 oxidation of RBCs mimics the morphological changes that occur in EPL erythrocytes with cells' aging
Please specify all components in formula 2. Theta is missing form the text. For AFM experts it might be familiar but for larger audience might not.
Answer: We thank the reviewer for this remark. In the revised version, all components in formula 2 are presented.
Minor spell check is required, like CNP is written instead NPC.
Answer: We thank the reviewer for the remark. Now it is corrected.

Reviewer 2 Report
The manuscript (Manuscript ID ijms-1684698) entitled “Morphometric and nanomechanical features of erythrocytes characteristic of early pregnancy loss” by Dr. Langari describe the use of an atomic force microscopy for evaluating the alteration of morphology and nanomechanics of erythrocytes from females with early pregnancy loss with unknown etiology. Main results indicate the presence of a time-dependent trend for the reduction of the cells' morphometric parameters (cells size and surface roughness) and the membrane elasticity which was much faster for early pregnancy loss compared to two control groups, comprising healthy pregnant and nonpregnant females. Despite several improvements are necessary to be made for improving the work, the manuscript is well written. The scientific writing style is adequate. Figures and tables are informative and detailed. In general, the ms will improve our knowledge on the employment of erythrocytes characteristics as biomarkers for early pregnancy loss evaluation. Considering the aforementioned aspects, I therefore recommend a minor revision. I have several suggestions for improving the manuscript:
General comments
1. In the results section. When necessary, I suggest including the non-significative p values (p>0.05) at the end of the sentences. For instance, page 3 last sentence. Similarly, significative p values should be included as well. For instance page 4, as well as many others. Please revise the entire text accordingly
2. P values and comparisons should be included in figure 2, 4, 5, 6 and 8
3. Conclusions should be moved at the end of the discussion
4. I suggest including more supporting references in the methods section, including 4.3, 4.4, 4.5 and 4.6 sub sections
Minor observations
Page 1, abstract, p values can be included within parenthesis
Page 1, additional causative factors commonly associated with pregnancy loss events comprise pathogenic infections (PMID: 30078192), immunological abnormalities (PMID: 27380207) as well as lifestyle factors, such as obesity (PMID: 33542866). This information and supporting references should be included
Page 1 Pregnancy loss characterized by an unknown etiology are referred to us idiopathic pregnancy loss. Please include this information
Page 2 (as well as other pages, such as page 3) instead of “women”, I suggest including “females”
Page 3, table 1, the two periods at the end of the table should be removed. In addition the NPC), PC and EPL annotation should also be specified in the table caption. The same comment can be done for table 2
Page 4 “Although the biconcave cells prevailed in the EPL group, their proportion was lower (53%) compared to the control groups” p value?
Page 7 “H2O2” should include underscore numbers
Page 9 “Erythrocytes are subject to constantly changing conditions in the bloodstream that can strongly affect their structural and biophysical properties” English should be improved
Page 10 “(Fig. 6).” These annotations should be removed from the discussion
Page 11 Ages (plus range) of females from control groups should be included
Author Response
Reviewer 2
The manuscript (Manuscript ID ijms-1684698) entitled “Morphometric and nanomechanical features of erythrocytes characteristic of early pregnancy loss” by Dr. Langari describe the use of an atomic force microscopy for evaluating the alteration of morphology and nanomechanics of erythrocytes from females with early pregnancy loss with unknown etiology. Main results indicate the presence of a time-dependent trend for the reduction of the cells' morphometric parameters (cells size and surface roughness) and the membrane elasticity which was much faster for early pregnancy loss compared to two control groups, comprising healthy pregnant and nonpregnant females. Despite several improvements are necessary to be made for improving the work, the manuscript is well written. The scientific writing style is adequate. Figures and tables are informative and detailed. In general, the ms will improve our knowledge on the employment of erythrocytes characteristics as biomarkers for early pregnancy loss evaluation. Considering the aforementioned aspects, I therefore recommend a minor revision. I have several suggestions for improving the manuscript:
Dear Reviewer,
We deeply appreciate your very insightful and constructive comments and recommendations. Hereby is a point-by-point reply to your comments. Changes made to the manuscript have been marked with track changes.
General comments
1. In the results section. When necessary, I suggest including the non-significative p values (p>0.05) at the end of the sentences. For instance, page 3 last sentence. Similarly, significative p values should be included as well. For instance page 4, as well as many others. Please revise the entire text accordingly
Answer 1: We thank the reviewer for the remark. Now the non- significative p values (p>0.05) and significative p values (p<0.05) are included.
- P values and comparisons should be included in figure 2, 4, 5, 6 and 8
Answer 2: The P-value is included in Figures 2, 4, 5, 6, and 8.
- Conclusions should be moved at the end of the discussion
Answer 3: According to the “Instructions for Authors” of the Journal, the “Conclusions” section should be after “Materials and Methods”
- I suggest including more supporting references in the methods section, including 4.3, 4.4, 4.5 and 4.6 sub sections
Answer 4: Now these sections are supported by more references.
Minor observations
Page 1, abstract, p values can be included within parenthesis
Answer: We thank the reviewer for the comment. p values are included within the parenthesis.
Page 1, additional causative factors commonly associated with pregnancy loss events comprise pathogenic infections (PMID: 30078192), immunological abnormalities (PMID: 27380207) as well as lifestyle factors, such as obesity (PMID: 33542866). This information and supporting references should be included
.Answer: We acknowledge this comment. The risk factors for early pregnancy loss were added with the corresponding references (Page 1, Introduction, lines 4-8)
Page 1 Pregnancy loss characterized by an unknown etiology are referred to us idiopathic pregnancy loss. Please include this information
Answer: The corresponding information was included in the text.(Page 1, Introduction, line 9)
Page 2 (as well as other pages, such as page 3) instead of “women”, I suggest including “females”
Answer: “women”, is replaced by “females” were necessary in the revised version, as according to obstetrics terminology the term Is women with prehnancy loss.
Page 3, table 1, the two periods at the end of the table should be removed. In addition the NPC), PC and EPL annotation should also be specified in the table caption. The same comment can be done for table 2
Page 4 “Although the biconcave cells prevailed in the EPL group, their proportion was lower (53%) compared to the control groups” p value?
Answer: This omission is fixed in the revised version.
Page 7 “H2O2” should include underscore numbers
Answer: We thank the reviewer for the remark! It is corrected.
Page 9 “Erythrocytes are subject to constantly changing conditions in the bloodstream that can strongly affect their structural and biophysical properties” English should be improved
Answer: We thank the reviewer for this remark! Now the sentence is improved.
Page 10 “(Fig. 6).” These annotations should be removed from the discussion
Answer: Now this is removed.
Page 11 Ages (plus range) of females from control groups should be included
Answer: Now, this omission is fixed.
